# Skin Extracellular Matrix Breakdown Following Paclitaxel Therapy in Patients with Chemotherapy-Induced Peripheral Neuropathy

**DOI:** 10.3390/cancers15164191

**Published:** 2023-08-21

**Authors:** Nathan P. Staff, Sybil C. Hrstka, Surendra Dasari, Enrico Capobianco, Sandra Rieger

**Affiliations:** 1Department of Neurology, Mayo Clinic, Rochester, MN 55905, USA; staff.nathan@mayo.edu (N.P.S.);; 2The Jackson Labs, Farmington, CT 06032, USA; 3Department of Biology, University of Miami, Coral Gables, FL 33146, USA; 4Sylvester Comprehensive Cancer Center, University of Miami Miller School of Medicine, Miami, FL 33136, USA

**Keywords:** CIPN, chemotherapy-induced peripheral neuropathy, paclitaxel, Taxol, skin, extracellular matrix, collagen, MMP-13, matrix metalloproteinase

## Abstract

**Simple Summary:**

Chemotherapy drugs like paclitaxel (Taxol) cause chemotherapy-induced peripheral neuropathy (CIPN) in the majority of patients. This condition of nerve degeneration is characterized by pain, tingling, temperature sensitivity, and numbness in the hands and feet and can persist even after chemotherapy treatment is completed. CIPN significantly affects the quality of life of cancer patients and can require the termination of chemotherapy if severe symptoms arise. Our prior analysis of the mechanisms of paclitaxel-induced peripheral neuropathy in zebrafish and rodent models revealed that the underlying cause of CIPN is a skin-specific breakdown of the extracellular matrix. In this study, we aimed to investigate the effects of paclitaxel in the skin of breast cancer patients with CIPN following paclitaxel therapy. We performed a skin punch biopsy on the distal leg of healthy control subjects and CIPN patients aged between 60 and 70 years. Although a similar nerve fiber density was present in the CIPN patient and control skin, the RNA sequencing of skin biopsies revealed significant gene expression changes in relation to the extracellular matrix, cytoskeleton, cell cycle regulation, and genes involved in the nervous system’s function. Furthermore, the expression of the extracellular matrix-degrading enzyme MMP-13 was elevated in the skin of CIPN patients, and also, when assessed with immunostaining, collagen breakdown, and basement membrane thinning were present at the ultrastructure. These findings indicate that extracellular matrix remodeling might also contribute to CIPN in patient skin following paclitaxel therapy.

**Abstract:**

The chemotherapeutic agent paclitaxel causes peripheral neuropathy, a dose-limiting side effect, in up to 68% of cancer patients. In this study, we investigated the impact of paclitaxel therapy on the skin of breast cancer patients with chemotherapy-induced peripheral neuropathy (CIPN), building upon previous findings in zebrafish and rodents. Comprehensive assessments, including neurological examinations and quality of life questionnaires, were conducted, followed by intraepidermal nerve fiber (IENF) density evaluations using skin punch biopsies. Additionally, RNA sequencing, immunostaining for Matrix-Metalloproteinase 13 (MMP-13), and transmission electron microscopy provided insights into molecular and ultrastructural changes in this skin. The results showed no significant difference in IENF density between the control and CIPN patients despite the presence of patient-reported CIPN symptoms. Nevertheless, the RNA sequencing and immunostaining on the skin revealed significantly upregulated MMP-13, which is known to play a key role in CIPN caused by paclitaxel therapy. Additionally, various genes involved in the regulation of the extracellular matrix, microtubules, cell cycle, and nervous system were significantly and differentially expressed. An ultrastructural examination of the skin showed changes in collagen and basement membrane structures. These findings highlight the presence of CIPN in the absence of IENF density changes and support the role of skin remodeling as a major contributor to CIPN.

## 1. Introduction

Taxanes, like other chemotherapeutic agents, cause several dose-limiting side effects, the most significant of which is peripheral neuropathy. According to a meta-analysis of patients with peripheral neuropathy induced by chemotherapy, up to 68 percent of patients treated with the widely used chemotherapeutic agent paclitaxel acquired peripheral neuropathy throughout the course of treatment [1]. This condition is characterized by sensory-dominant symptoms, including paresthesia, allodynia, hyperalgesia, and acute pain [2], which manifest in stocking and glove distribution [3]. Peripheral neuropathy has a significant impact on cancer treatment because patients with severe symptoms require dose reduction or treatment discontinuation, lowering their chance of survival, whereas cancer survivors experience a significant decrease in their quality of life due to permanent nerve damage as a result of treatment [4]. In all, chemotherapy-induced peripheral neuropathy (CIPN) has a significant impact on the outcome of cancer treatment, survival, and the patient’s quality of life.

Over the past decade, there has been intensive research focusing on the mechanisms underlying paclitaxel-induced peripheral neuropathy, and several key pathways and processes have been found to be altered, including intracellular calcium signaling [5,6,7], axonal transport [8,9,10], the mitochondrial function [8,11,12], neuropeptide secretion [13,14,15], and immune cell interactions with sensory axons [16,17,18]. 

Besides these neuron-intrinsic changes, animal studies have shown the significant role of the skin in the induction of paclitaxel-induced peripheral neuropathy [19,20], which is consistent with the earliest signs of neuropathy detected in intraepidermal nerve fibers [21,22,23]. Our research in zebrafish previously established that paclitaxel therapy causes considerable modifications and damage to epidermal keratinocytes [20,24]. We identified that epidermal MMP-13 upregulation with paclitaxel treatment causes extracellular matrix degradation and axon degeneration and showed that MMP-13 is a conserved mediator of paclitaxel-induced peripheral neuropathy since its pharmacological inhibition prevents cutaneous axon degeneration in zebrafish, mouse, and rat models [19,20].

The purpose of this study was to determine if and to what extent paclitaxel therapy also induces changes in the skin of breast cancer patients with CIPN. The findings support our previous observations and show a significant upregulation of MMP-13 in CIPN patients who received their first CIPN diagnosis at 5, 31, and 35 weeks prior to skin biopsy. Other changes include a reduction in the dermal collagen fibril diameter and basement membrane thinning when examined at the ultrastructure. RNAseq further determined significantly downregulated genes involved in extracellular matrix regulation and upregulated genes with a role in microtubule and cell cycle regulation, as well as nervous system function. Together, these findings demonstrate persistent changes in the skin of patients, which could contribute to CIPN.

## 2. Materials and Methods

### 2.1. Patient Clinimetric and Biosample Collection

Patient recruitment and IRB approval have been reported elsewhere [24]. In brief, three female subjects with breast cancer who received standard adjuvant paclitaxel neurotoxic chemotherapy (12 weekly infusions of 80 mg/m^2^) and who developed neuropathic symptoms were identified and enrolled along with 3 age and sex-matched healthy volunteers without neuropathy, all aged between 60–70 years (as shown in Table 1). 

Neurological history, examination, Neuropathy Impairment Scores the in lower limbs (NIS-LL), and QLQ-CIPN20 quality of life questionnaires were completed in all subjects. NIS-LL is a composite score of strength, sensation loss, and reduced deep tendon reflexes in the lower extremities [25]. The QLQ-CIPN20 questionnaire, which has been validated to detect the presence of CIPN [26], consists of 20 items that are designed to assess various dimensions of CIPN-related symptoms and their impact on daily functioning and quality of life. It covers a wide range of domains, including sensory symptoms (such as tingling, numbness, and pain), motor symptoms (such as weakness and difficulty with coordination), autonomic symptoms (such as sweating abnormalities and gastrointestinal disturbances), and functional impairments (such as difficulties with fine motor skills, walking, and activities of daily living). Participants were asked to rate the severity and frequency of these symptoms over the past week on a scale ranging from 1 (not at all) to 4 (very much). The questionnaire also included items that assessed the participants’ perception of the overall impact of CIPN on their quality of life, including its effects on their physical, emotional, and social well-being. It provides valuable insights into the participants’ subjective experiences and helps quantify the burden of CIPN symptoms on their day-to-day lives.

Two 3 mm skin punch biopsies were performed 10 cm proximal to the lateral malleolus on the distal leg, as described previously [27]. The biopsies were carried out following the local injection of 2% lidocaine with epinephrine anesthesia under sterile conditions. The biopsies were processed for immunostaining, transmission electron microscopy, and RNA sequencing.

### 2.2. RNA Sequencing Study

The RNA sequencing study has been reported elsewhere [24]. The dataset is deposited under GSE228633. For this study, the dataset was processed to account for low expression genes using the DESeq2 version 1.34.0 default method with *p*-value adjustment for multiple testing using the Benjamini–Hochberg procedure. TMM (the trimmed mean of M values) was based on log-fold changes between samples for medium-expression genes. The method did not employ any gene length normalization to identify differentially expressed (DE) genes between samples, and thus, it was assumed that the gene length was constant across samples. The resultant significant (*p* adj value < 0.05) and differentially expressed genes were used in iDEP [28] for clustering, biological pathway, and cellular component predictions. MMP and collagen genes were selected from the differentially expressed gene list and queried in STRING [29] to generate network interactions. STRING analysis was used to identify networks of interest. For the mitochondrial analysis, all genes (logFC) and the selected FDR were compared to MitoCarta3.0 [30], which produced a list of genes that were used for both network analysis and final annotations. The Excel file shows the overlapping MitoCarta genes and pathway annotations. For collagen-interacting gene identifications, DAVID was used to generate gene annotations. All results are shown in the Appendix A.

### 2.3. Immunostaining

Skin biopsies processed for MMP-13 staining were fixed in 4% paraformaldehyde (PFA)/1×Phosphate-buffered saline (PBS) for 16 h at 4 °C while gently rocking. The sections were paraffin-embedded and sectioned at a 4 µm thickness following standard procedures. Prior to immunostaining, the sections were deparaffinized by incubation in 100% xylene 2 × 3 min, followed by 1:1 Xylene:100% ethanol for 3 min, 100% ethanol 2 × 3 min, 95% ethanol/1×PBS for 3 min, 70% ethanol for 3 min, and 50% ethanol for 3 min. This was followed by rinsing the sections in running cold tap water for 1 min using Mason jars. Antigen retrieval was not necessary. Following a 5 min incubation in 1×PBS + 0.1% Tween-20 (PBST) and rocking at room temperature, the skin sections were permeabilized in 1×PBS + 0.1% Triton X-100 for 5 min at room temperature while rocking. This was followed by incubation in a blocking buffer (1×PBST + 5%BSA) for 30 min at room temperature with rocking. Next, 200 µL of the antibody solution containing the rabbit anti-human MMP-13 antibody (1:100, ProteinTech Group, Inc. (Rosemont, IL, USA) Cat. No. 18165-1-AP) in a blocking buffer was added to each section, carefully covered with a long coverslip over the entire slide, and sections were incubated at 4 °C in a humidity box (closed lid plastic box with moist Kimwipes at the bottom, and 5 mL serum pipettes broken into halves on top, onto which the slides were placed to avoid contact with the Kimwipes). The next morning, slides were transferred into Mason jars, coverslips were removed, and sections were washed 4 × 15 min in 1×PBST at room temperature with rocking. This was followed by incubation in a secondary antibody (1:1000, goat anti-mouse Cy3, Abcam (Boston, MA, USA), Cat. No. ab97035) and 1:10,000 Hoechst 33342 (ThermoFisher Scientific (Waltham, MA, USA), USA Cat. No. 62249, 20 mM solution) in a blocking buffer for 1 h at room temperature in a humidity box wrapped in aluminum foil to avoid photobleaching. The sections were washed 4 × 15 min in 1×PBST at room temperature, rocking in covered Mason jars, and subsequently treated with a VectaShield mounting medium (Vector Laboratories, Newark, CA, USA). 

Skin biopsies processed for PGP9.5 staining, a marker for sensory nerve endings in the epidermis and quantification of IENF density, were performed at the Mayo Clinic Peripheral Nerve Laboratory according to standard clinical practice procedures (regulated by Clinical Laboratory Improvements Amendments—CLIA) as described previously [27].

### 2.4. Transmission Electron Microscopy

For transmission electron microscopy (TEM), skin biopsies were first fixed in Trump’s fixative consisting of 4% formaldehyde and 1% glutaraldehyde in PBS (pH 7.2), followed by post-fixation in 1% osmium tetroxide and staining en bloc with 2% uranyl acetate. The grids with epidermis were viewed using an FEI Technai 12 transmission electron microscope at 100 kV (Fei, Inc., Hillsboro, OR, USA), equipped with a digital CCD camera (Advanced Microscopy Techniques, Danvers, MA, USA). The grids with dermis were viewed at 80 kV in a JEOL JEM-1400 transmission electron microscope, and images were captured with an AMT BioSprint digital camera.

### 2.5. Image and Statistical Analyses

Immunofluorescence imaging was performed on the LSM880 Airyscan confocal microscope (Zeiss, Oberkochen, Germany) using a 20× air objective and zoom between 1× and 2.5×. Sections in varying stack sizes using 1 µm step sizes were recorded to capture the full thickness of the skin section using a pixel resolution of 1024 × 1024. Scanning was performed at a 7 s scan speed and averaging factor 8. 

Images were processed and analyzed in Imaris 9.5.1 (Bitplane, Schlieren, Switzerland) and Fiji 2.9.0. Following quantification, graphs were generated, and statistics were performed in Prism 9 (GraphPad) software. For publication images, projected stacks were saved as .tif files, followed by processing in Photoshop 2023 to assemble the figures. Schematics were prepared in BioRender (https://www.biorender.com/, accessed on 16 July 2023).

IENF quantifications were conducted as described in [26]. All other quantifications were performed in Fiji. Nuclei and MMP-13 staining were imaged in identical settings. Fluorescence was quantified by measuring the mean background intensity in each image and subtracting the intensity from each mean fluorescence intensity of a region of interest. The background was first subtracted from each object’s fluorescence measurement, and subsequently, the mean ratio of MMP-13 fluorescence was divided by the mean nuclear fluorescence intensity and multiplied by 1000 to normalize for staining intensity variations. Statistical analyses were performed using Student’s *t*-test for the comparison of two groups (control vs. CIPN patients). The lengths and widths of mitochondria were measured in Fiji. The length–width ratios were calculated as a measure of morphology. For mitochondrial intermembrane space measurements, at least 5 lines per mitochondrion were drawn, spanning the circumference of the mitochondrion for width measurements. The data were statistically analyzed using Prism 10.

## 3. Results

### 3.1. Intraepidermal Nerve Fiber Quantifications

Comprehensive assessments were performed to evaluate the participants’ neurological status and the impact of CIPN on their quality of life. This included gathering a detailed neurological history, conducting neurological examinations that were quantified with NIS-LL, and administering the QLQ-CIPN20 quality of life questionnaire, which has been validated for its ability to detect the presence and severity of CIPN [26,31]. Subjects in the paclitaxel-treated group had signs and symptoms of sensory neuropathy, as is seen in paclitaxel-induced CIPN [32] (Table 1).

In addition to the QLQ-CIPN20 questionnaire and NIS-LL, objective measures were obtained by assessing the IENF density through 3 mm skin punch biopsies taken at 10 cm proximal to the lateral malleolus on the distal leg, as described previously [27]. The IENF density ranged from 14.9 (70 years) to 7.6 (64 and 65 years) in the control group and 9.6 to 5.6 in the CIPN group (CTRL: 10.03 ± 4.2 versus CIPN: 7.93 ± 2.3, SD), which did not significantly differ due to the low IENF density in two of the control subjects (Table 1). The IENF density was within normal limits given the age of the participants and based on laboratory normative values [27]. The lack of significance between the control and the CIPN group despite the reported CIPN symptoms suggested other changes that could not be directly attributed to axon degeneration, such as modifications to the skin as observed in animals [19,20], which could contribute to the development of CIPN. 

### 3.2. RNA Sequencing

Considering prior research findings that show the presence of skin damage as an early indicator of CIPN in animal models caused by MMP-13 upregulation, we investigated whether there were gene expression differences in the skin of control and CIPN subjects despite the absence of significantly different IENF densities. For this, we analyzed our RNA sequencing dataset (GSE228633), in which we compared skin biopsies of three control and three CIPN subjects (Figure 1a). Of the 2227 genes that were significantly and differentially expressed (adjusted *p*-value < 0.05) in patients compared with the control subjects, 1574 genes were upregulated, and 653 genes were downregulated (Figure 1b). Given that we identified MMP-13 as a major player in the etiology of CIPN in zebrafish and rodent models, we assessed the expression levels of significantly and differentially expressed *MMPs*. Although the expression levels of *MMP13* (ENSG00000137745) were relatively low overall and especially low in the control skin (CTRL005: 0.2018 TPM; CTRL006: 0.11 TPM; CTRL007: 0.25 TPM versus CIPN001: 1.69 TPM; CIPN002: 1.68 TPM; CIPN003: 1.76 TPM), its log2 fold-change (FC) was 3.28 when comparing the CIPN and control group. Compared with other significantly and differentially expressed MMPs (FC: *MMP16*: 2.238; *MMP24*: 1.94 TPM; *MMP2*: *−*1.70 TPM), this was the highest log2-FC (Figure 1c).

Because MMPs are known regulators of collagen, and collagen breakdown products in the serum have been found to be a hallmark of various diseases [33,34,35,36,37,38,39], we also assessed the expression levels of all significantly and differentially regulated collagen genes in the skin. We detected 5 upregulated collagen genes (highest to lowest upregulation: *COL11A1*, *COL22A1*, *COL2A1*, *COL8A1*, *COL26A1*) and 6 downregulated genes (highest to the lowest downregulation: *COL1A2*, *COL18A1*, *COL5A3*, *COL6A2*, *COL6A3*, *COL15A1*). Of the upregulated collagens, COL2A1 has been previously demonstrated to be a substrate for MMP-13 in osteoarthritis (OA) [40] and has been suggested as a disease biomarker [41] (*COL2A1*: CTRL005: 10.14 TPM; CTRL006: 15.30 TPM; CTRL007: 8.34 TPM versus CIPN001: 38.58 TPM; CIPN002: 90.260; CIPN003: 58.19 TPM; logFC: 2.42). The downregulated collagens included *COL1A2*, *COL6A3*, and *COL6A2*, of which COL1A2 is a likely substrate for MMP-13 [42]. COL1A2 is present mostly in connective tissue fibroblasts of the dermis. Interestingly, the silencing of *COL1A2* via methylation is common in melanoma cell lines and tumors [43], suggesting that paclitaxel treatment could induce cell changes that are also present in cancerous cells. In summary, there is evidence suggesting abnormal collagen regulation in the skin. However, further investigations are needed to determine whether skin collagen breakdown products can be utilized as biomarkers for predicting the onset of CIPN.

To extract more meaningful information from the RNAseq dataset, we next conducted a k-means clustering analysis using iDEP [28] (Figure 1d). K-means is a powerful technique that can obtain valuable insights into cellular processes and biological phenomena by grouping genes or samples into distinct clusters based on their expression profiles. It is a useful tool for exploratory data analysis that can aid in discovering biological patterns or identifying subsets of genes with similar regulatory mechanisms or functional roles. This analysis identified 8 clusters in the patient dataset of which 6 contained upregulated and 2 downregulated genes. One of the downregulated gene clusters from the k-means analysis (cluster A) harbored genes that are implicated in extracellular matrix changes when queried for the gene ontology (GO) the term *“*Cellular Component*”*. Specific subclusters that contained most of the genes were in pathways, such as “Extracellular region”, “Extracellular space”, “Collagen-containing extracellular matrix”, “Extracellular organelle”, “Extracellular exosome”, “Extracellular vesicle”, “External encapsulating structure”, and “Extracellular matrix”. This finding is consistent with the role of MMP-13-dependent ECM breakdown in paclitaxel*-*exposed skin. Since two of the patients were diagnosed with CIPN 31 and 35 weeks prior to the skin biopsy, it indicates that extracellular changes persist for an extended period. Intriguingly, cluster D harbored upregulated genes that were involved in the microtubule and cell cycle function, which is consistent with a role for paclitaxel in these processes in cancer cells, suggesting that similar processes are activated in healthy skin. 

Two additional clusters (G and H) harbored genes that are involved in the nervous system function (Figure 1d), which is rather surprising and indicates that these genes either regulate sensory neurons within the skin or they represent localized mRNAs within nerve endings. It is possible that these transcripts are located in Schwann cells, which are involved in neuronal signaling. For instance, a key signaling factor involved in myelination is *Neuregulin-1 (NRG1)* [44,45,46,47], which was significantly upregulated (log-FC: 1.85, padj: 0.01). NRG1 has also been implicated in skin development and function, including epidermal proliferation, differentiation, and stratification, hair follicle and sweat gland development, and wound healing, in addition to its role in sensory axon branching [48,49,50,51,52]. Its specific role in CIPN is unknown, and it remains to be investigated. Other genes in these clusters can, for example, be subcategorized into “synaptic membrane” and included: *CNTN1*, *CDH10*, *SYNDIG1*, *SYP*, *HTR2A*, *CSPG5*, *GRIA3*, *DLG4*, *SNAP25*, *SYT11*, *EPHB2*, *EPHA7*, *FAIM2*, *PCDH8*, *ADAM10*, *SLC16A3*, *CACNG8*, *NLGN4X*, *CASK*, *CDH8*, *PTPRO*, *GRID2*, *GRIA4*, *GRIA1*, *SORCS3*, *ATP2B2*, *DGKI*, *SHANK1*, *GRIK2*, *ADCY1*, *CACNG2*, *NLGN1*, *CNIH2*, *NRXN1*, *CNTN2*, *NLGN3*, *LRRTM3*, and *GRIN2B*. Out of these genes, nine have not been characterized with respect to skin-related functions (CNTN1, SYNDIG1, CSPG5, FAIM2, SLC16A3, SORCS3, DGKI, CNIH2, LRRTM3). Since sensory nerve endings in the skin do not form synapses [53], processes that mimic the synaptic function, such as specific interactions between sensory neurons and Merkel cells [54]*,* can be perturbed and contribute to the axonal deficits experienced by CIPN patients. For instance, syndecan-interacting protein 1 (*SYNDIG1*) participates in the organization of neuronal synapses and is part of a larger protein family that also includes antiviral proteins, which restrict fusions between the host and viral membranes [55]. Sortilin-related VPS10 domain-containing receptor 3 (*SORCS3*) has been implicated in protein trafficking and sorting in neurons, potentially impacting synaptic function and neuronal development [56]. These genes might have comparable functions in damaged skin to mediate responses to insults, such as paclitaxel therapy to promote axonal regeneration or function. It is also possible that detected skin-specific transcripts reflect mRNA present in the nerve endings of skin-innervating sensory neurons, which may be destined for localized mRNA translation [57,58]. To further determine collagen and MMP networks and identify possible functional implications, we queried the differentially expressed collagen, MMP, and Tissue inhibitors of matrix metalloproteinase (TIMP) genes in STRING. This showed various clusters, one of which was collagen degradation, in which MMP13 was implicated among other MMPs (Figure 1e). We further used DAVID to identify additional collagen regulators that were differentially expressed in our dataset (Appendix A). MMP13 was associated with 32 other genes that were annotated for extracellular matrix and collagen fibril organization, which were significantly and differentially expressed in the CIPN patient’s skin. Because mitochondria have been implicated in neurodegenerative diseases, such as CIPN [59,60,61,62,63], but no study has examined skin mitochondria in this context, we performed this analysis and assessed whether we could detect altered mitochondrial gene expression in the CIPN patients by comparing our significantly and differentially expressed gene set to MitoCarta3.0. This database harbors 1136 genes spanning seven broad categories related to mitochondria. This analysis showed that 34 genes were implicated in mitochondrial function, several of which were involved in Ox/Phos and metabolic regulation (Appendix A). One intriguing observation is that the gene expression changes were still present after 31 and 35 weeks following the first CIPN diagnosis, suggesting that paclitaxel induces persistent skin damage. Although the epidermis undergoes a renewal process every 28–30 days, the dermis renews infrequently. Thus, either irreversible changes are induced in epidermal cells, such as at the epigenetic level or the primary damage occurs in the dermis.

### 3.3. MMP-13 Expression Is Increased in the Skin of CIPN Patients

Given the strong increase in *MMP13* expression in CIPN patient skin, we next analyzed MMP-13 protein expression using immunofluorescence staining. MMP-13 staining was observed in all skin layers in the control subjects and CIPN patients (Figure 2). The most striking difference was in the dermis, where the staining appeared brighter and distinct in the CIPN patient’s skin. The epidermis also showed MMP-13 expression differences, but the signal was much weaker overall when compared to dermal expression. Because of this, we quantified only the brightest cells in each skin compartment, including the stratum spinosum and granulosum (SS + SG), stratum basale (SB), and dermis (Figure 2a). The data were normalized in each scan and combined for each patient group (2*–*4 sections per patient). In this analysis, MMP-13 fluorescence was significantly increased in the full-thickness of CIPN skin compared with control subjects (Figure 2b,c). The mean fluorescence intensity difference between the controls and CIPN patients for SS + SG was 1.71*-*fold, whereas the SB difference was slightly lower (1.59*-*fold). The highest difference in MMP-13 expression was observed in the dermis, where MMP-13 expression levels were twice as high in the patient skin compared with the control skin (2.05-fold difference). Interestingly, Hoechst 33342 staining revealed that the nuclei in the patient group contained large dark regions, which were not observed in the control group, and were stained and imaged under identical conditions. This phenotype resembled zebrafish nuclei following paclitaxel treatment, where detyrosinated (stabilized) microtubules pierced through the nuclei in the epidermal keratinocytes. These regions appear darker when observed with Hoechst33342 staining [24]. We also showed that nuclear piercing correlated with the increased activation of Nox-dependent reactive oxygen species (ROS) formation upstream of *mmp13* expression. Therefore, a similar mechanism could upregulate *MMP13* in human skin, which remains to be investigated. 

### 3.4. Ultrastructural Analysis of the Skin

We next sought to analyze the skin ultrastructure in order to assess changes in IENFs and collagen, which is consistent with the MMP and collagen gene expression changes. We first analyzed epidermal sections and found that in the control epidermis, unmyelinated cutaneous axons were visible and often located in bundles of 2*–*3 axons (Figure 3a,b). Most of the axons were visible in abundance in the basal (SB) layer. Axons in a comparable region in one CIPN patient (35 weeks after diagnosis), on the other hand, were absent, although large gaps remained visible (Figure 3a,c). This finding is consistent with previous findings in zebrafish treated with paclitaxel, where the epidermis also showed large gaps in which cutaneous sensory axons are normally present [19]. Thus, the surrounding ECM where IENFs are previously present appears to be maintained in a similar organization over longer time periods. 

We previously uncovered that the treatment of zebrafish with paclitaxel leads to a discontinuous basement membrane [19]. The basement membrane, or dermal-epidermal junction, is situated beneath the basal (SB) layer and separates the epidermis from the dermis. This compartment is rich in collagen IV, VII, and XVII [64]. Although we did not detect changes in these collagens at the gene expression level in CIPN patients, the upregulated collagen, COL2A1, is known to be present in the skin basement membrane [65] and is, furthermore, regulated by MMP-13 [40]. It was, therefore, interesting to determine whether we could observe basement membrane changes in the CIPN patient’s skin. Indeed, we found regional basement membrane degradation at the ultrastructure (Figure 3c). Further quantifications revealed a significantly reduced overall basement membrane diameter in the patient group compared with the control subject’s skin (Figure 3d). This suggests that increased MMP-dependent collagen degradation can underlie the basement membrane thinning, whereby increased *COL2A1* expression in our RNAseq data set could potentially represent a compensatory upregulation or be attributed to another cell population. Previously, we showed in zebrafish that mitochondria in epidermal keratinocytes were abnormally condensed following paclitaxel treatment. We, therefore, analyzed whether morphological changes were evident in the epidermal mitochondria of CIPN patients. We quantified the length-width ratio as a measure of organelle shape but did not find significant differences between the control and CIPN subjects (Figure 3e). We did observe a significant reduction in the width of the mitochondrial intermembrane space in CIPN patients, which could indicate reduced metabolic activity (Figure 3f). Consistent with changes in mitochondrial gene expression, this reduction could suggest an altered mitochondrial function in the patients.

We next analyzed the collagen ultrastructure in the dermis, given the vast amount of collagen in this skin compartment and significant gene expression differences in CIPN patients. The dermis, positioned between the epidermis and the subcutaneous fat layer, is a crucial layer of mesenchymal connective tissue. It consists of two main regions: the papillary dermis, located beneath the epidermis, which is characterized by loose and highly vascular connective tissue, and the deeper reticular layer, which constitutes the majority of the dermis and is composed of dense connective tissue. Within the dermis, collagen emerges as the predominant type of extracellular matrix (ECM) protein, providing structural support and integrity. MMP-13 has been shown to split native collagen fibrils into individual protein strands, in contrast to MMP-1, for instance, which induces breakages and kinks in collagen fibrils, leading to a disorganized pattern [66]. In support of the predominant involvement of MMP-13, our observations revealed no significant collagen disorganization, as evident by a similar appearance in both the control and CIPN patient papillary dermis (Figure 4a,b). However, cross-sectioned collagen fibrils appeared grainy compared with those in the control dermis, and reticular collagen fibrils had a significantly reduced diameter in CIPN patients (Figure 4b,c). This reduction was not uniform among individual fibrils; however, it was consistent with the degradation of specific sites along the fibril. 

## 4. Discussion

We have provided data on a small cohort of patients with breast cancer who developed sensory neuropathy following paclitaxel treatment. The limitations of our study were the small number of enrolled subjects (*n* = 3 for each group) and their homogeneity with respect to sex and ethnicity, which included only white female subjects. In addition, one patient varied with respect to the time of CIPN diagnosis (5 weeks versus 31 and 35 weeks prior). Future studies will need to include a larger cohort with increased diversity and additional matched time points of diagnosis. Nevertheless, we found a large number of significantly and differentially expressed genes in the CIPN patient cohort, and the k-means cluster analysis identified a gene cluster consistent with our previous findings for the role of ECM degradation in CIPN. Additionally, a cluster related to microtubule regulation and cell cycle gene expression was identified, which is consistent with the role of paclitaxel in altering cell cycle regulation in cancer cells. Most surprising was that the CIPN patients showed two large clusters of upregulated genes involved in the nervous system function. In conclusion, paclitaxel treatment induces robust changes in skin-specific gene expression, which can influence sensory neuron function without inducing axon degeneration. Since these profound molecular changes could be detected even in a small patient cohort, some of these genes may be useful as CIPN biomarkers and could be collected by performing a superficial skin biopsy, which remains to be investigated.

Interestingly, we found that the number of nerve endings in the CIPN patient cohort was comparable to the control group despite the presence of a significant neuropathy score in this group. Since CIPN was diagnosed several weeks prior to the skin biopsy, the nerve endings could have regenerated to some degree without regaining their function. Such an assumption is consistent with findings that knockout mice for the pro-degenerative factor, *Sarm1*, showed preserved axons but with reduced functionality [67]. Thus, axonal function may not be intimately linked with structural presence. However, in our study, it is more likely that the disconnect between insignificant IENF density and CIPN presence stemmed from the advanced age of the participants. In two of the three control subjects, IENF densities were very low, although within the normal range. However, for some individuals, this low baseline may be normal, but others may have higher baseline levels. Our patients may, therefore, have true CIPN but did not fall into this category due to statistical estimates. Longitudinal studies of these individuals would have been necessary beforehand to determine their baseline levels. 

In addition to these confounding factors, IENF density differences may have been more subtle in the leg, which we biopsied, as compared with the palms and soles, where degeneration is most prominent [68]. Our studies further suggest that changes in skin gene expression might have also contributed to CIPN given the changes in ECM regulatory genes and those mediating cell cycle and cytoskeletal organization as well as nervous system function, which is consistent with our previous findings in zebrafish and mice [20,24]. Therefore, animal models, such as zebrafish and rodents, can be reliably used to study CIPN mechanisms. 

## 5. Conclusions

In conclusion, our study on a small cohort of patients with breast cancer who developed paclitaxel-induced sensory neuropathy, combined with our previous studies in zebrafish and rodent models, provides valuable insights into potential CIPN mechanisms in humans. Despite the limitations of small sample size and homogeneity within the cohorts, our findings support the idea that CIPN is not solely determined by the structural presence of axons but that gene expression changes in the skin of CIPN patients may have contributed to the symptoms. Interestingly, we also showed that significant molecular changes remain even after many weeks following CIPN diagnosis, indicating that these persistent changes could be a valuable assessment tool for CIPN. Although further research with larger and more diverse patient cohorts is warranted to support this study, we have previously validated these findings mechanistically in zebrafish. Thus, this combined knowledge may aid in the development of improved treatment strategies for the prevention, early detection, and management of CIPN.

## Figures and Tables

**Figure 1 cancers-15-04191-f001:**
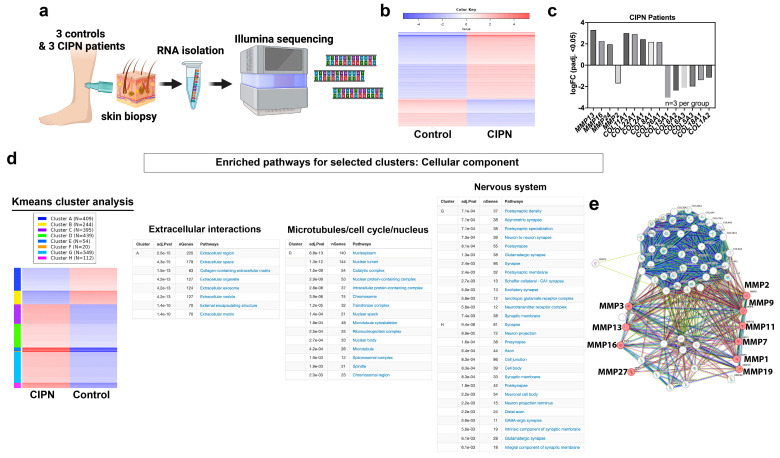
RNA sequencing reveals extracellular matrix breakdown. (**a**) Schematic of RNAseq analysis using human skin biopsies from three healthy subjects and three CIPN patients. (**b**) Heatmap representing log fold-change differences (n=3 per group). Blue represents downregulated genes and red upregulated genes. (**c**) Expression data for significant MMP and collagen genes. (**d**) K-means analysis identified eight clusters of which four are shown. Cluster A harbors downregulated genes involved in extracellular matrix regulation whereas cluster D is implicated in cell cycle, nucleus, and microtubule-related pathways. Genes in clusters G and H have been implicated in nervous system function. (**e**) STRING analysis of all differentially expressed collagen, *MMP* and *TIMP* genes shows *MMP13*, *MMP3, MMP7, MMP16, MMP27, MMP2, MMP9, MMP11, MMP1*, and *MMP19* are involved in collagen degradation processes (colored dots).

**Figure 2 cancers-15-04191-f002:**
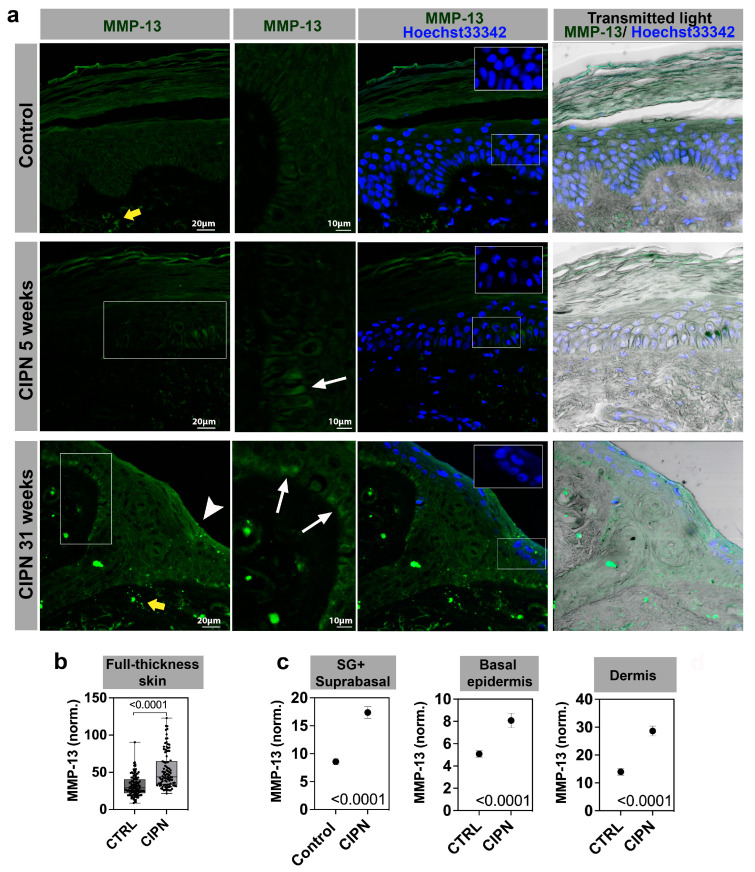
MMP-13 immunofluorescence staining in different skin compartments. Stratum spinosum and granulosum, SS + SG; stratum basale, SB; and dermis. (**a**) MMP-13 antibody staining in control (top panel) and two CIPN patients (middle panel at 5 weeks, and lower panel at 31 weeks post CIPN diagnosis). White boxes depict zoomed-in regions shown in the second column. MMP-13 positive cells are present in the basal (SB) layer (white arrows). The white arrowhead in the lower left panel points to MMP-13 positive cells in the SS + SG layer. The yellow arrows depict increased dermal expression in the CIPN skin. Insets with Hoechst33342 stain show darker nuclei in the CIPN patient skin compared with the control subject skin. The scale in unlabeled images is identical to the panel on the left. (**b**) Quantification of MMP-13 expression in full-thickness skin. (**c**) Quantification of MMP-13 expression in sub-compartments (SG + SS vs. SB vs. dermis).

**Figure 3 cancers-15-04191-f003:**
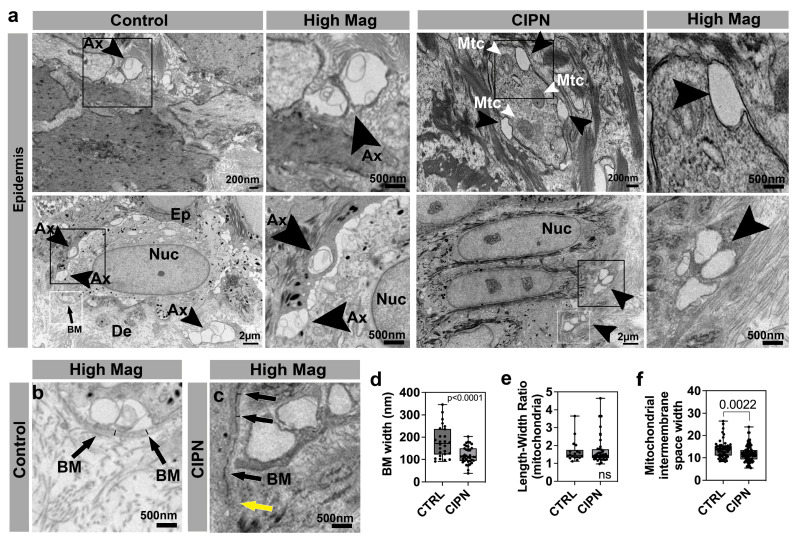
Epidermal changes in CIPN patients. (**a**) Intraepidermal nerve endings (arrowheads) are abundant in control subject skin whereas gaps are present in the CIPN patient skin. (**b**) High magnification of control skin shown in (**a**) depicting the basement membrane (arrows) and width demarcated by the black lines. (**c**) High magnification of CIPN patient skin shown in (**a**) depicting the basement membrane (black arrows) and width demarcated by the black lines. The yellow arrow points to the less dense region of the basement membrane. (**d**) Quantification of the basement membrane width (white boxes in a) shows a reduction in the CIPN patient skin. (**e**) Length–width ratio (LWR) of mitochondria in epidermal keratinocytes shows no differences between the control subjects and CIPN patients. (**f**) Mitochondrial intermembrane space reduction in CIPN patients. Abbreviations: Ax: Axon, BM: Basement membrane, De: Dermis, Ep: Epidermis, Mtc: Mitochondria, Nuc: Nucleus.

**Figure 4 cancers-15-04191-f004:**
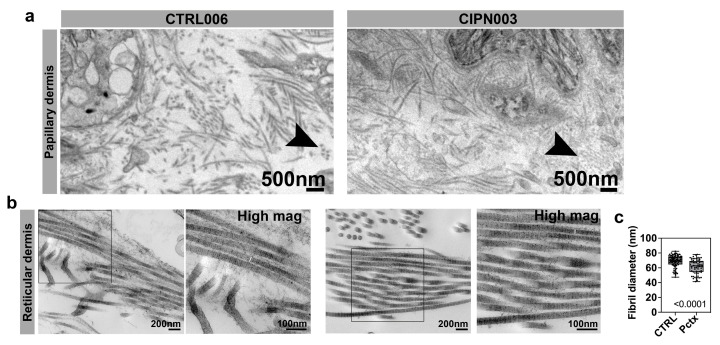
Dermal collagen abnormalities in CIPN patients. (**a**) Collagen fibrils in the papillary dermis of a 65*-*year*-*old control subject and a 60-year-old patient who was diagnosed with CIPN 35 weeks prior. Cross-sectioned collagen fibrils in the CIPN patient appear granular and small compared with the control subject (black arrowhead). (**b**) The right panel for each group shows the higher magnification of the boxed area in the left panel. There is no difference in the reticular collagen organization, but the fibrils appear thinner in the CIPN patient, as indicated by white lines in the magnified images. (**c**) The mean fibril diameter for reticular collagen is significantly reduced in CIPN patients.

**Table 1 cancers-15-04191-t001:** Patient information and neuropathy scores. Sex, age, and enrollment dates of 3 healthy control and 3 CIPN patients who received paclitaxel therapy are shown. Skin biopsies were taken at the indicated weeks following CIPN diagnosis. The QLQ-CIPN 20 scores are represented as the total score, sensory-specific or motor-specific scores. NIS-LL and epidermal nerve fiber density (IENFs) are also shown.

Study ID	Treatment	Sex	Age at Consent	Date of Enrollment	Weeks form Neuropathy to Consent	NIS-LL	CIPN20 Score (Total: 19–76)	CIPN20 Score (Sensory: 9–36)	CIPN20 Score (Sensory: 3–12)	IENF
001	Pctx	F	70	4/17/18	35	2	31	22	3	8.9
002	Pctx	F	70	6/7/18	31.43	12	34	22	4	5.3
003	Pctx	F	60	9/18/18	4.86	4	25	16	3	9.6
005	CTRL	F	70	9/21/21	n/a	0	19	9	3	14.9
006	CTRL	F	65	10/5/21	n/a	0	19	9	3	7.6
007	CTRL	F	64	11/16/21	n/a	0	19	9	3	7.6
**Average (Pctx vs. CTRL)**			**67/66**		**23.76333333**	**6**	**30/19**	**20/9**	**3.3/3.0**	**7.9/10.3**

## Data Availability

All data is included in this manuscript. Additional raw data can be requested from the authors. The RNA sequencing data was deposited under GSE228633.

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
