# Peer review of "Skin Extracellular Matrix Breakdown Following Paclitaxel Therapy in Patients with Chemotherapy-Induced Peripheral Neuropathy"

_cancers, 2023, doi:10.3390/cancers15164191_

Round 1

Reviewer 1 Report

This article presents compelling findings on the effects of paclitaxel therapy in the skin of CIPN patients. The study's novelty lies in uncovering paclitaxel's previously unknown long-lasting impact on the skin, which may contribute to CIPN. This manuscript also establishes a novel aspect of CIPN etiology in humans, supported by animal studies, revealing how skin ECM breakdown plays a role in this condition.

Before publication, several questions have emerged that warrant addressing:

1) Concerning ultrastructure analysis, do abnormalities in mitochondria, cell shape, and/or nuclei appear visible in the epidermis, as supported by other animal studies and, in part, by the RNAseq data?

2) It would be valuable to assess the RNAseq data for potential abnormalities in the expression of mitochondrial regulatory genes.

3) To provide a comprehensive view of gene expression networks in the skin of CIPN patients, could the authors generate a gene network involving collagen, MMPs, and genes regulating collagen and MMPs? Such an analysis could enhance our understanding of the underlying mechanisms in humans.

Overall, this study offers valuable insights into CIPN and its potential implications for patients undergoing paclitaxel therapy. Addressing the raised questions will further strengthen the manuscript and contribute to its significance in the field.

Author Response

We thank the reviewer for his/her valuable comments:

1) Concerning ultrastructure analysis, do abnormalities in mitochondria, cell shape, and/or nuclei appear visible in the epidermis, as supported by other animal studies and, in part, by the RNAseq data?

This is a great suggestion. We analyzed mitochondria and did not observe morphological changes, as assessed by measuring the length-width ratio (LWR). We however found a significant reduction in the intermembrane space in keratinocyte mitochondria of CIPN patients. On page 10, lines 399-407, we inserted the following text: “Previously, we showed in zebrafish that mitochondria in epidermal keratinocytes were abnormally condensed following paclitaxel treatment. We therefore analyzed whether morphological changes are evident in epidermal mitochondria of CIPN patients. We quantified the length-width ratio as a measure of organelle shape but did not find significant differences between the control and CIPN subjects. We did observe a significant reduction in the width of the mitochondrial intermembrane space in CIPN patients, which may indicate reduced metabolic activity. However, we did not find mitochondrial gene expression differences, suggesting that this reduction does not significantly alter mitochondrial function in the patients."

The data is shown in Figure 3e and f.

We did not observe altered nuclear shapes as determined by LWR in epidermal keratinocytes and opted to exclude this data.

2) It would be valuable to assess the RNAseq data for potential abnormalities in the expression of mitochondrial regulatory genes.

 We used MitCarta3.0 to compare our dataset with known human mitochondrial genes and found a list containing 34 significantly differentially expressed genes in our dataset, which overlap with this database. We have added the detailed information as Supplemental table 2. We performed a similar analysis for collagen-interacting genes using DAVID. This information is shown in the same Supplemental file.

3) To provide a comprehensive view of gene expression networks in the skin of CIPN patients, could the authors generate a gene network involving collagen, MMPs, and genes regulating collagen and MMPs? Such an analysis could enhance our understanding of the underlying mechanisms in humans.

This is an excellent idea. We performed this analysis and added the data as Figure 1e. In the manuscript on page 7 lines 312-31, we wrote: “To further determine collagen and MMP networks and identify possible functional implications, we queried the differentially expressed collagen, MMP and Tissue inhibitors of matrix metalloproteinases (TIMP) genes in STRING.  This showed various clusters, one of which was collagen degradation, in which MMP13 was implicated among other MMPs (Figure 1e).”

Reviewer 2 Report

Authors presented an article on a very interesting paper about Skin extracellular matrix breakdown following paclitaxel therapy in patients with chemotherapy-induced peripheral neuropathy. The study aimed to investigate the effects of paclitaxel in the skin of breast cancer 21 patients with CIPN following paclitaxel therapy.
Very interesting results showed that extracellular matrix remodeling may also contribute to CIPN in patient skin following paclitaxel therapy.

No major concerns, I would just like to underline the role of muscle strenght in older patients (especially in cipn) and overall the role of neuropathic pain [10.3390/diagnostics11040613]. This is particular relevant for the role of rehabilitation in those patients.

Author Response

We agree with the reviewer on the importance of strength and neuropathic pain in CIPN and its role in rehabilitation for those patients.  We think that our data contribute importantly to the recognition of this issue.

Reviewer 3 Report

This is a very well-documented paper that builds on the authors' previous research on the mechanism of chemotherapy-induced peripheral neuropathy involved in the response to paclitaxel. This is an important area of research, since the neuropathy induced by chemotherapeutic drugs can lead to cessation of treatment in severe cases, shortening the patient’s life.

The authors identify two areas of weakness:  The small size of the study (3 patients as experimental subjects and 3 patients as controls), and the limitation of the study to white females. The great strength of this paper is that the experimental design and technical approaches required have all been worked out, so upscaling the study should present no problems. This should permit larger cohorts, as well as the inclusion of gender and race as variables.

Author Response

We greatly appreciate the positive review.